# Effective Unidirectional Wetting of Liquids on Multi-Gradient, Bio-Inspired Surfaces Fabricated by 3D Printing and Surface Modification

**DOI:** 10.3390/polym16131874

**Published:** 2024-06-30

**Authors:** Che-Ni Hsu, Ngoc Phuong Uyen Mai, Haw-Kai Chang, Po-Yu Chen

**Affiliations:** 1Department of Materials Science and Engineering, National Tsing Hua University, Hsinchu 300044, Taiwan; ninihsuu@gmail.com (C.-N.H.); mainpuyen@gmail.com (N.P.U.M.); hkai1616@gmail.com (H.-K.C.); 2Instrumentation Center, National Tsing Hua University, Hsinchu 300044, Taiwan

**Keywords:** gradient wettability, anisotropic wetting, surface modifications

## Abstract

The movement of liquid droplets on the energy gradient surface has attracted extensive attention inspired by biological features in nature, such as the periodic spindle-shaped nodes in spider silks and conical-like barbs of cacti, and the structure–property–function relationship of multifunctional gradient surfaces. In this study, a series of specific patterns are fabricated with 3D printing technology, followed by modification via the atmospheric pressure plasma treatment and liquid phase chemical deposition, resulting in enhancing the ability of water droplets of 5 μL to travel 18.47 mm on a horizontal plane and 22.75 mm against gravity at up to a 20° tilting angle. Additionally, analysis techniques have been employed, including a contact angle analyzer, ESCA, and a laser confocal microscope to evaluate the sample performance. This work could further be applied to many applications related to microfluidic devices, drug delivery and water/fog collection.

## 1. Introduction

Controlled self-propelling liquid transport is of considerable interest for specific applications, such as self-cleaning, microfluidic devices, water/fog collection and condensing equipment [1,2,3,4,5]. This field has been well studied by previous researchers in general [6,7,8,9,10,11], while Laplace pressure gradient force and wettability gradient from biological properties particularly have drawn much attention as the main driving forces for unidirectional wetting [6,12,13,14,15,16,17,18,19,20]. For instance, the spine of cacti accumulates fog in a humid environment using their barbs. Due to the hierarchical cone-like shape, the droplet is driven to move from the tip to the base [12,16]. The Nepenthes peristome possesses a unique surface with overlapping drop-shaped grooves, generating the oriented capillarity mechanism to transfer droplets from the inner to the outer sides [17,18,19]. The periodic spindle-knots and joints of spider silks display a series of semitransparent puffs composed of cylindrical silk thread, which is naturally hydrophilic. The spindle-knots are more hydrophilic than the joints, causing fog to condense on the silk and then move from the joint to the knot [20,21]. Moreover, the unidirectional wetting behavior of desert beetles has been uncovered with the combination of hydrophilic bumps and hydrophobic channels on the exoskeleton, the droplet accumulates on the bumps and rolls down the beetle back when it overcomes the capillary force [21,22].

Previous studies have been exploiting the natural traits to regulate the liquid spreading combining different geometrical anisotropy and wettable gradients [8,12,15,23,24,25,26,27,28,29]. Deng et al. used the anodic oxidation method to formulate the wettable gradient surface, which was then coated with paraffin wax to create a hydrophobic background [15]. A wedge pattern was removed to generate a Laplace pressure gradient on the wettable gradient surface, leading to the water droplets self-transport unidirectionally 2.4 mm under the given volume of 5 μL. Zhang et al. used an electrospinning technique with a high-speed collection drum to synthesize an aligned PET/CHI fibrous surface, and the resulting parallel structure improves the wetting behaviors [28]. Then, the lifting–dissolution method was applied to the fibrous surface to achieve gradient wettability. The traveling distance of the water droplet on the gradient fibrous surface was 12.3 mm with 5 μL in volume. Zhu et al. took advantage of the beetles’ structure, which synthesized a superhydrophilic background through alkaline oxidation to the smooth copper mesh [29]. In a sequence of dipping into the mixer of n-hexane and n-octadecyl thiol/ethanol to achieve the superhydrophobic circular patterns. The efficiency in water capturing was outperformed with 1107.5 mgh^−1^cm^−2^.

Our research is motivated by the unique characteristics of cactus and spider silk as mentioned above, we further investigate the anisotropic wetting with an array of triangular prism grooves through 3D printing technology integrating with surface modification, including plasma treatment and polymer grafting, as shown in Figure 1. The wedge-shaped pattern with the Laplace pressure gradient (LG) would drive the droplet from the tip (high LG) to the base of the triangle (less LG) on a wettable gradient surface. Moreover, the multi-gradient surface is sufficient to manipulate the droplet to spread against gravity at varying tilting angles. Besides that, the surface profiles of the patterns were analyzed by laser confocal microscopy, contact angle analyzer and electron spectroscopy for chemical analysis (ESCA). We expect that such a simple and economical design could be potential work for further research and practical applications for unidirectional wetting.

## 2. Materials and Methods

### 2.1. Preparation of 3D-Printed Triangular Prism Pattern

The design parameters of the structural samples are presented in Figure 2. A single groove was created by the edges of two adjacent wedge-shaped protrusions. The length of the groove was set to be 5 mm, and the tilt angle of 20° was selected based on the apex angle of spindle-knots and barbs, which were revealed to be 19° and 19.5°, respectively [6,12]. This study focused primarily on two factors that affect droplet spreading, including the spacing between two emerged patterns (S) or the width of the groove’s narrow tip, and the thickness of the protrusion (T). In particular, this structure created a Laplace pressure gradient as a driving force which directed the droplets to flow unidirectionally. In addition to the triangular prism pattern, the tetrahedron patterns were designed and printed and their wetting properties were evaluated. Appendix A depicts the tetrahedron design and its dimensions, featuring a groove consisting of two triangles in the base and wall. All samples were printed by a Formlabs2 SLA 3D printer (Formlabs, Somerville, MA, USA) with liquid ray resin as the printed slurry. The printer was equipped with a 405 nm ultraviolet laser, and a layer thickness was printed at 25 μm. The models were designed by SolidWorks (Dassault Systemes A.A., Waltham, MA, USA), and the stereolithographic file was exported into Preform software (Formlabs, London, UK). After printing, the printed samples were thoroughly rinsed with isopropanol in an ultrasonic machine for 15 min and then put into the ultraviolet oven to cure and remove the residual moisture of the samples.

### 2.2. Preparation of PEGMA-Grafted 3D Printed ABS-like Samples

In conjunction with the Laplace pressure gradient, we introduced the gradient wettability to enhance the unidirectional, self-driving wetting behavior of droplets. The detailed experimental process involved two main steps: the non-pattern substrates were treated by atmospheric pressure plasma with a fixed working distance (non-inclination). On the other hand, atmospheric pressure plasma treatment was conducted at an inclined angle of 20° to achieve the wettable gradient surface. The static contact angles at different locations of the plasma treated samples were measured.

Plasma was introduced to activate the surface with plenty of free radicals to perform temporary hydrophilicity. Hence, these plasma-treated sample surfaces were subsequently immersed in PEGMA solution (Mn~440, Alfa Aesar, Ward Hill, MA, USA) at 60° for 24 h to generate a stable wettability. Finally, the grafted samples were rinsed thoroughly with deionized water in an ultrasonic machine for 10 min to eliminate the unreacted PEGMA molecules adhered to the samples and then dried in the oven at 60° for an hour and prepared for the following measurements.

### 2.3. Water Flow Behavior Observation

The experiment was set up with a contact angle analyzer (First Ten Angstroms 1000, Portsmouth, NH, USA) equipped with a charge-coupled device (CCD) and a digital camera to record the wetting process. All experiments were conducted at room temperature (24–26 °C) and 50–70% relative humidity. Deionized water droplets of 2 μL were deposited on flat, non-patterned samples to observe the static contact angle while blue-colored deionized water droplets of 5, 10, and 15 μL were dropped on samples with bio-inspired patterns under horizontal (0°) and inclined (10°, 20°, 30°) conditions to examine the dynamic, unidirectional wetting behavior as well as the capability to flow against gravity. Moreover, the CCD from the contact angle apparatus was also utilized to measure the time-dependent durability of hydrophilicity on the modified flat substrates. The traveling distance and velocity of the water droplets were further analyzed by the ImageJ software (National Institutes of Health, Bethesda, MD, USA). At least three individual measurements were performed on each modified sample. At least three samples were fabricated and tested for each group.

## 3. Results and Discussion

### 3.1. Gradient Wettability on Flat ABS-like Substrates

#### 3.1.1. Static Angle Measurement and Dynamic Wetting Behavior

The static contact angle was measured from 2 μL water droplets on a non-pattern ABS-like substrate with different surface treatments, as shown in Figure 3. The intrinsic contact angle of the 3D-printed ABS-like substrate was 100.1° ± 1.4°. After being treated by atmospheric pressure (AP) plasma at a working distance of 3.4 mm, the contact angle decreased considerably to 12.2° ± 5.8°. The long-term hydrophilicity of the plasma-treated surface was examined within 30 days in Figure 4, each point corresponding to a droplet of 2 μL. Unfortunately, the plasma-treated sample reached a 95.1° ± 2.2° contact angle, which seemed to recover to its initial state in the first 7 days, because of the effect of plasma to generate active species, which spark diffusing from the bulk surface toward the inner side and cover the thermodynamically unstable surface [30]. To address that drawback, the plasma-treated one is further immersed in PEGMA solution, resulting in a contact angle that continues reducing to 11.1° ± 0.8°. Moreover, the hydrophilicity of PEGMA-grafted ABS-like substrate could maintain at least 30 days at 12.3° ± 2.7° on average, successfully sustaining a hydrophilic state. The long-term durability of hydrophilicity is reckoned with the stable bonding between functional groups of PEGMA and plasma-treated ABS-like surface, which is proved by ESCA in the next section.

For further experiments to generate gradient wettability, the flat ABS-like substrate is altered surface by atmospheric pressure plasma at an inclined angle of 20°. The contact angle plotted with their respective working distances is shown in Figure 5, which goes along with the CCD images to illustrate the water droplet behavior on these positions. In general, the trend in contact angle increases gradually from 9.5° to 67.7°, relating to the working distance in the range of 1.9–17.2 mm, which means droplets deposited on the gradient surface can propagate in the same direction (Figure 5a). In the following procedure, PEGMA is grafted onto the plasma-treated sample in order to improve the efficiency and durability of hydrophilicity. Figure 5b illustrates the increase in the contact angle from 12.1° to 57.1° with a rising working distance from 3.4 mm to 15.4 mm. In terms of dynamic wetting behavior, the droplets drop at positions of 23 mm, 30 mm, and 44 mm in Figure 6. The droplet embarks on spreading at the hydrophilic end and then wets unidirectionally towards the hydrophobic end, with traveling distances of 3.59 mm, 4.83 mm, and 6.01 mm, respectively.

#### 3.1.2. Chemical Composition Analysis

The ESCA technique is utilized to verify the change in chemical components of modified surfaces. In accordance with the spectra in Figure 7, the peak fitted narrow scan C1s spectrum of the untreated ABS-like surface is compared to that of PEGMA-grafted ones with two specific working distances. Regarding the XPS C1s spectra of untreated surfaces, three peaks are examined, including C-C and C-Si bonds at 284.6 eV, C-O or C≡N bonds at the peak of 286.1 eV, and R-O-C=O carboxylic groups or imide groups at the peak of 288.8 eV. Before being grafted with PEGMA solution, plasma treatment at working distances of 3.6 mm and 14.4 mm is conducted separately to observe the alternation in terms of chemical composition. The peak intensity at 286.1 eV conspicuously increases, which is caused by the formation of C-O bonds of PEGMA. Additionally, the existence of C-O and C≡N at the intensity of 286.1 eV is associated with reducing working distance, pointing out that the ABS-like surface is placed near the plasma jet improving the bonding formation for the grafting process.

### 3.2. Dynamic Wetting Behavior on Multi-Gradient 3D-Printed Triangular Prism Patterns

The triangular prism groove obsesses a unique way to drive droplets from the high-Laplace-pressure to the lower-Laplace-pressure regions, which is discovered through barbs of cactus and spider silk in Figure 1 [12,20]. According to the aforementioned results, the sample with triangular prism grooves is treated with AP plasma at an inclined angle of 20° and thenceforth PEGMA grafting to synthesize multi-gradient surfaces. Consequently, two driving forces exist on the pattern to support the droplet for self-transport. The design parameters and laser confocal images of the pattern are shown in Figure 8, the accuracy in thickness and length of the printed ABS-like samples reaches over 98% in comparison to the model in Figure 9.

The optimal design for manipulating the water self-transport has been verified by adjusting the thickness (T) and the spacing (S) of the modified samples in Figure 10, which shows a series of frames depicting their difference in wetting behavior. For the samples with a thickness of 0.1 mm and varied spacing (0–0.5 mm), the droplets only spread 5 mm in distance at the less hydrophilic position because the groove is not well-defined, resulting in the Laplace pressure being unfunctional. The samples with a thickness of 0.3–0.5 mm contrarily show a decrease in traveling distance with increasing spacing between adjacent patterns (S = 0.3–0.5 mm) due to the Laplace pressure force being inversely proportional to the triangle radius [31]. As a result, the longest traveling distance of 26.7 mm is observed on the pattern with 0.3 mm in thickness and 0.3 mm in spacing, which is selected as an optimized dimension for the following experiments.

Furthermore, the wetting process on the modified pattern with an optimized parameter is entered into detail in Figure 11 the volume droplet of 5 μL is deposited on a groove of the hydrophobic side, and the water droplet flows 18.47 mm along the groove column within 6 s. Surprisingly, the result is much higher than that of the untreated (in the hydrophobic state) and uniform hydrophilic 3D-printed triangular prism pattern with measurements of 2.6 mm and 8.8 mm, as shown in Figure 12. In comparison to the triangular prism, the tetrahedron-patterned sample may have three primary factors: the Laplace pressure force on the base and the wall of the groove, the gravity generated on the upper surface of the extruded pattern, and the hydrophilic surface. However, unidirectional wetting could not happen on untreated or modified tetrahedron patterns, as shown in Appendix A. The droplets were pinned on the groove and could not spread further.

In the following process, the significant change in droplet velocity during the process is analyzed in Figure 11b. Owing to the principle of Laplace pressure force, the droplet accelerates near the triangle’s tip region and then steadily retards as it reaches the rear region. Similarly, the wetting behavior is observed as an increase in droplet volume to 10 μL and 15 μL, as shown in Figure 13 and Appendix A, respectively. For the case of 10 μL, the droplet fills another three grooves longer than that of 5 μL in volume. Therefore, the traveling distance extends from 18.47 mm to 26.51 mm. A distance of 43.97 mm is witnessed with 15 μL in water volume and almost approaches the other end of the hydrophilic region.

In further investigation, the multi-gradient pattern carries out the anti-gravity trial, and the stage of the contact angle apparatus is set up, titling at 10°, 20°, 30°, and 40°. Figure 14 and Appendix A show a summary of the results that water droplets with 5 μL, 10 μL, and 15 μL wet against gravity at various tilting angles. Regarding the tilting angle of 10° in Figure 14a, the droplet climbs up to 17.57 mm with 5 μL in volume, and the traveling distance is 0.95 times shorter than the case without inclination. The volume of a droplet rises to 10 μL and 15 μL, which allows it to flow longer—25.92 mm and 40.48 mm, respectively. It is seen that the self-transport efficiency can be retained at over 92% in comparison to the non-inclination cases (18.47 mm for 5 μL, 26.51 mm for 10 μL and 43.97 mm for 15 μL). As the titling angle is set to be 20° in Figure 14b, the droplets of 5 μL, 10 μL, and 15 μL correspond to 17.15 mm, 22.75 mm, and 36.34 mm in distance against gravity, respectively. Consequently, the efficiency of self-transport is realized at 83% compared to the cases of the horizontal plane, except for the droplet volume of 5 μL, which can be kept at 93% efficiency. In terms of 30° in Figure 14c, the anti-gravity process is recorded with an insignificant value of 6.39 mm for 5 μL and 12.53 mm for 10 μL in traveling distance. Unfortunately, the multi-gradient force of the triangular prism pattern is no longer against gravity when 15 μL in volume is loaded on the pattern, causing the drop to partially run down. In summary, the anti-gravity water flow is confirmed on the multi-gradient surface; the integration of the Laplace pressure gradient and the wettable gradient is sufficient to drive the droplet running uphill along the channel from 10° to 30° inclination.

## 4. Conclusions

The combination of gradient wettability and Laplace pressure gradient improves the self-transport unidirectionally without any external energy. In this research, we fabricated a substrate with a series of triangular prism grooves through 3D printing, followed by a simple and controllable two-step process to generate a multi-gradient surface. The wettable gradient surface was elucidated by measuring the static contact angle at different positions on the substrate and inspecting the change in chemical composition. Two primary results are mentioned in this study: First, the traveling distance of 18.47 mm for 5 μL on the multi-gradient surface is 2.1 times, 5.1 times, and 7.1 times higher than the hydrophilic structural surface (8.8 mm with Laplace pressure gradient only), flat gradient surface (3.59 mm with wettable gradient only), and hydrophobic structural surface (2.6 mm with Laplace pressure gradient only), respectively. Secondly, water droplets of 5 μL to 10 μL in volume can run uphill up to a 30° tilting angle, while the droplet with 15 μL could run against gravity up to an angle of 20°. Such a design offers great potential in transporting small water droplets against gravity, which could be further applied in microfluidic drainage tubes for fog or rain collection.

## Figures and Tables

**Figure 1 polymers-16-01874-f001:**
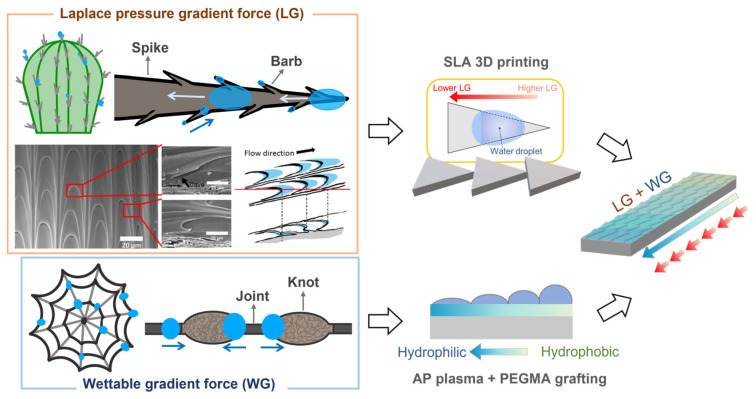
The schematic illustrates the multi-gradient triangular prism pattern. Reproduced with permission. Copyright 2015, Chiao-Peng Hsu et al., Springer Nature [17].

**Figure 2 polymers-16-01874-f002:**
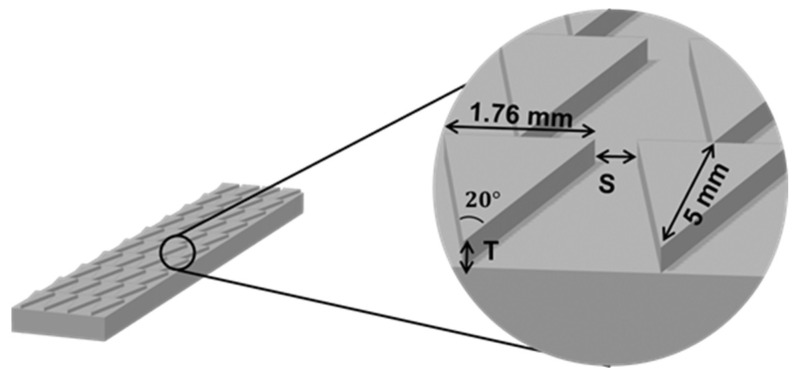
Design parameters of the ABS-like samples with triangular prism patterns.

**Figure 3 polymers-16-01874-f003:**
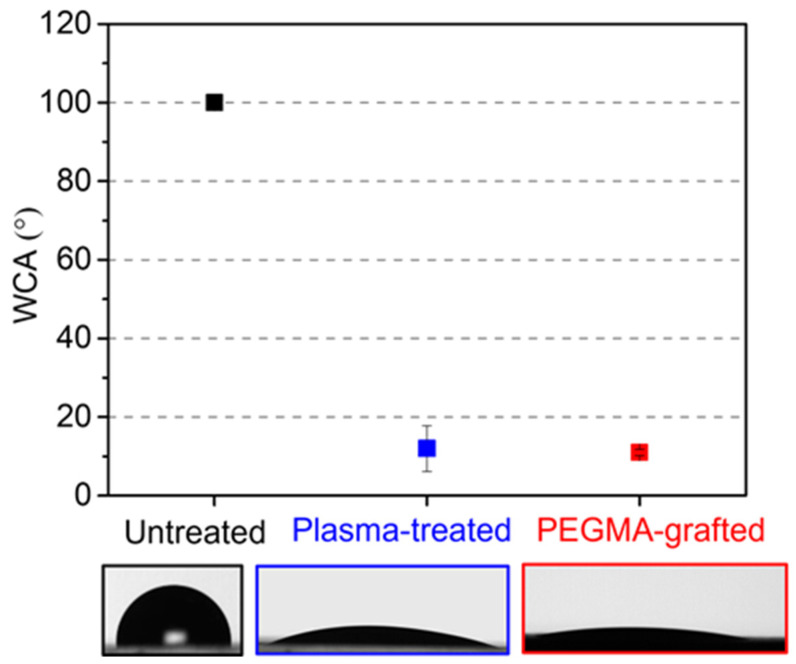
The plot of the water contact angle versus untreated, plasma-treated, and PEGMA-grafted samples corresponds with the CCD images.

**Figure 4 polymers-16-01874-f004:**
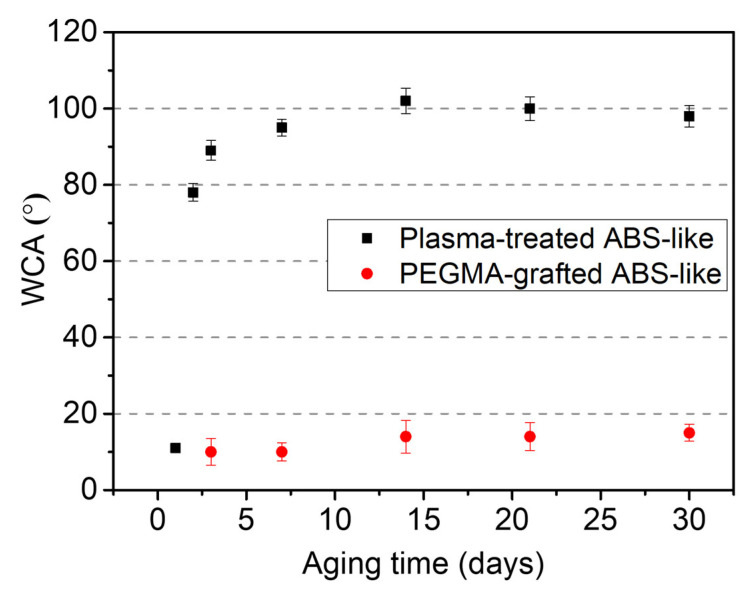
The plot of water contact angle versus aging time.

**Figure 5 polymers-16-01874-f005:**
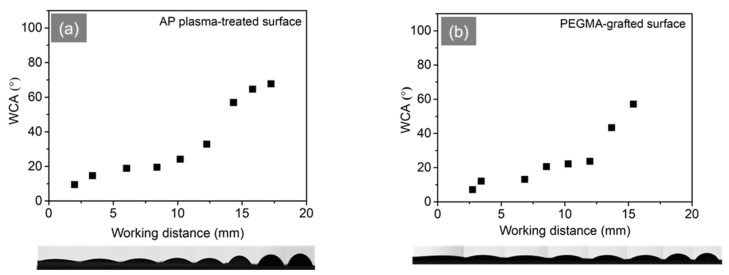
Plot of water contact angles versus the working distance corresponding to the gradient wettability from CCD images. Flat ABS-like samples were prepared and treated by (**a**) AP plasma at an inclined angle of 20°, (**b**) followed by PEGMA-grafted. The water droplet is 2 μL.

**Figure 6 polymers-16-01874-f006:**
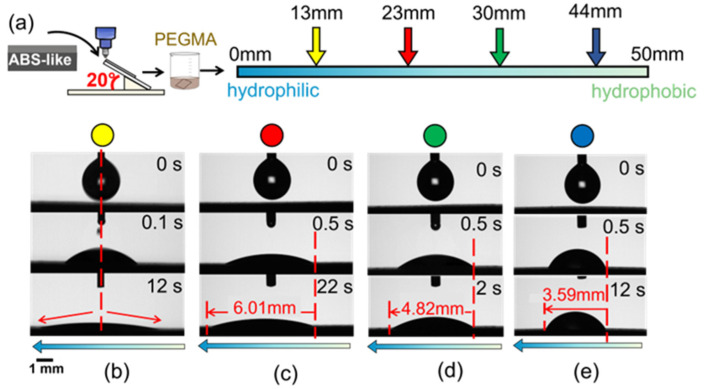
(**a**) The schematic diagram illustrates the surface modification process and the location corresponding to the deposition of droplets on the PEGMA-grafted ABS-like surface. The unidirectional wetting behavior at the position of (**b**) 13 mm; (**c**) 23 mm; (**d**) 30 mm; (**e**) 44 mm. The blue arrow indicates the wettable gradient force. The water droplet volume is 5 μL.

**Figure 7 polymers-16-01874-f007:**
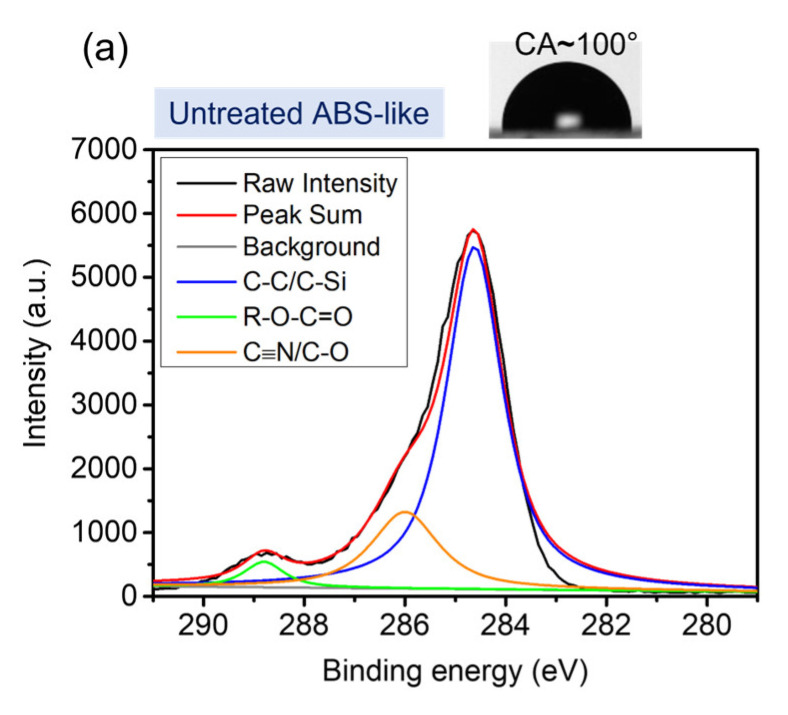
ESCA spectrum of C1s for (**a**) untreated ABS-like substrate; PEGMA-grafted ABS-like substrate at a working distance of (**b**) 3.6 mm; (**c**) 14.4 mm.

**Figure 8 polymers-16-01874-f008:**
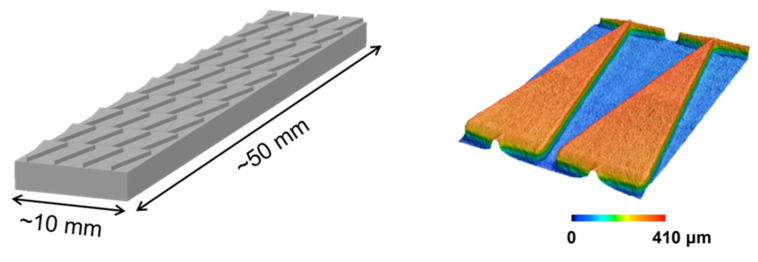
3D printed ABS-like with triangular prism array and the laser confocal images (color image) showing the structure contour difference design structure (grey image).

**Figure 9 polymers-16-01874-f009:**
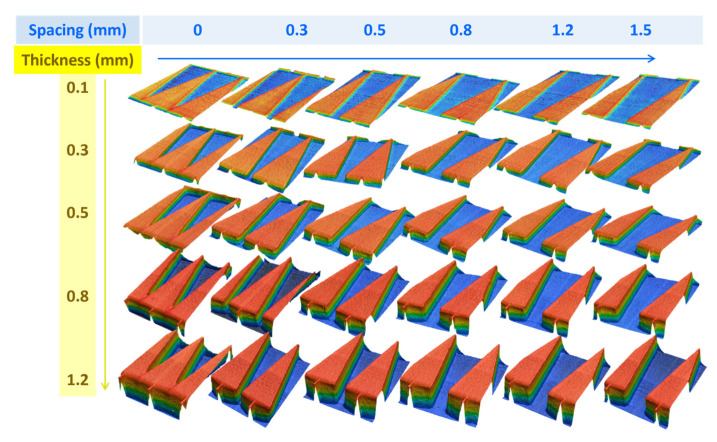
Laser confocal images show the actual spacing (0, 0.1, 0.3, 0.5, 0.8, 1.2 mm from the left to the right) and the actual thickness (0.1, 0.3, 0.5, 0.8, 1.2 mm from the top to the bottom) of the printed samples.

**Figure 10 polymers-16-01874-f010:**
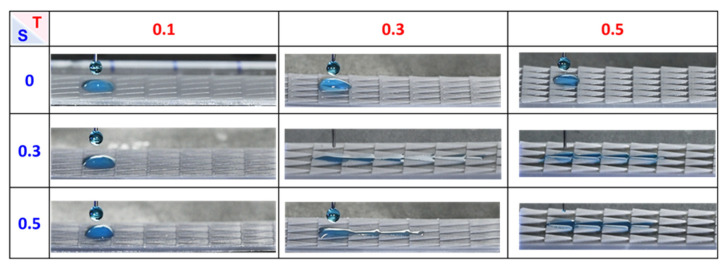
The traveling distance of a droplet on the different samples corresponding to each CCD image. T is defined as the thickness of the triangular prism groove. S is defined as the spacing between each pattern. The volume of a water droplet is 10 μL.

**Figure 11 polymers-16-01874-f011:**
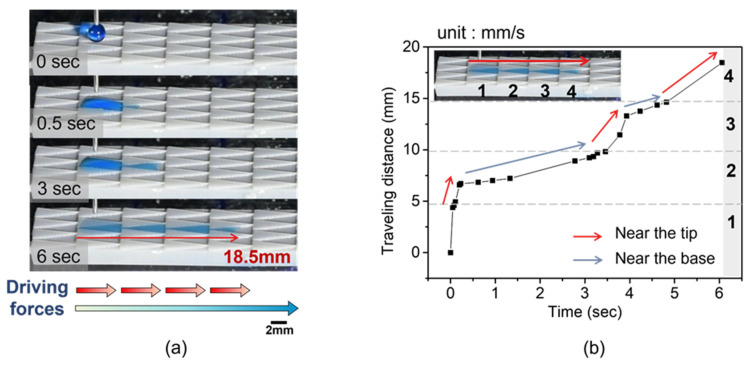
(**a**) The unidirectional spreading behavior on the multi-gradient surface. The blue and red arrows show the trend of water flow driven by the wettable gradient force and Laplace pressure gradient force, respectively. The volume of a water droplet is 5 μL; (**b**) the droplet of the traveling distance versus time on the multi-gradient surface. The numbers 1–4 designate groove order.

**Figure 12 polymers-16-01874-f012:**
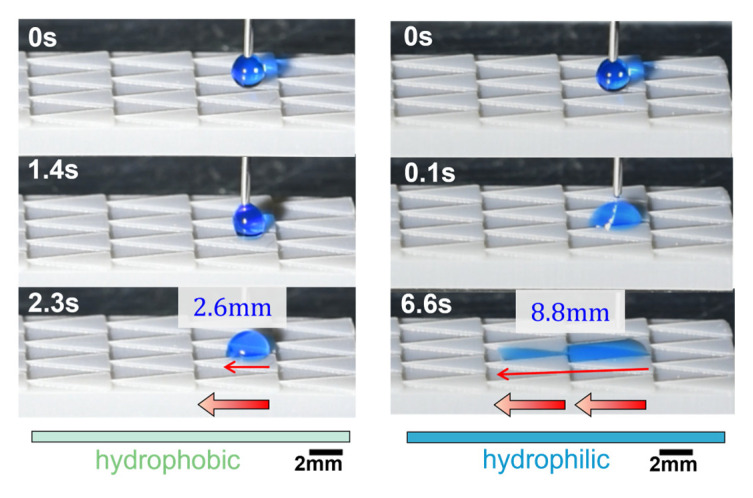
The dynamic wetting behavior on 3D printed triangular prism samples. The lighter blue and darker blue indicate the hydrophobic and hydrophilic surfaces, respectively. The red arrow indicates the direction of water flow. The droplet volume is 5 μL.

**Figure 13 polymers-16-01874-f013:**
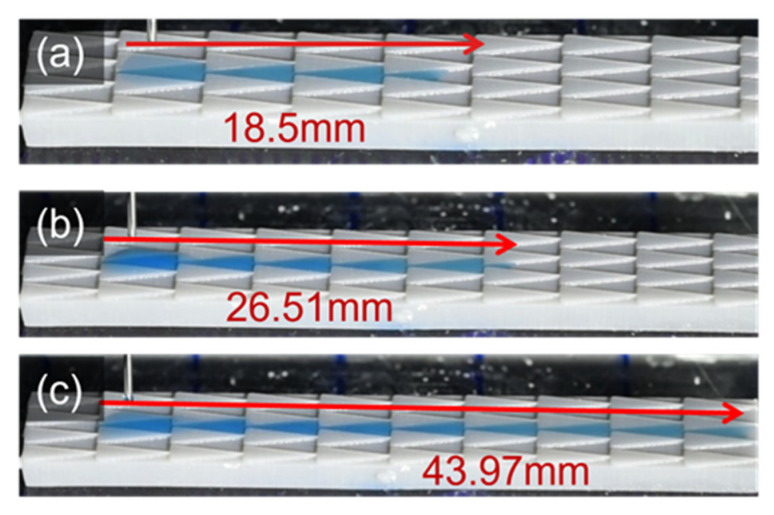
The unidirectional wetting behavior on the multi-gradient surface. The volume of a droplet is (**a**) 5 μL; (**b**) 10 μL; (**c**) 15 μL.

**Figure 14 polymers-16-01874-f014:**
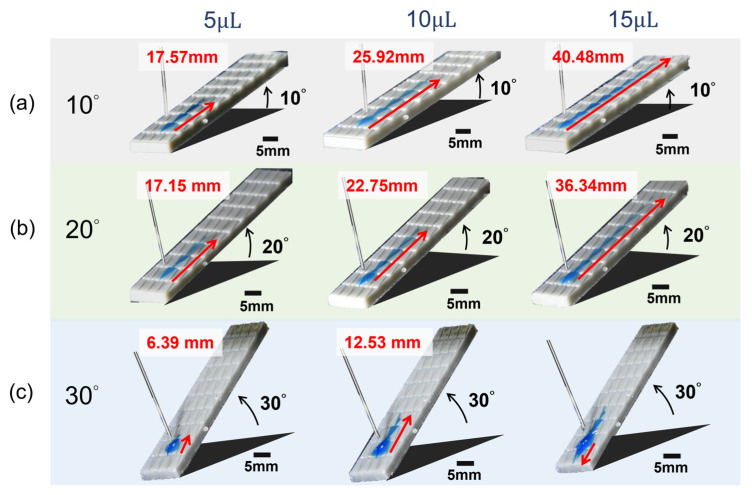
The anti-gravity water flows on the multi-gradient surface. The tilting angle is set to be (**a**) 10°; (**b**) 20°; (**c**) 30°, with droplet volumes of 5 μL, 10 μL, and 15 μL.

## Data Availability

The original contributions presented in the study are included in the article/Appendix A. Further inquiries can be directed to the corresponding authors.

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
