# Peer review of "Effective Unidirectional Wetting of Liquids on Multi-Gradient, Bio-Inspired Surfaces Fabricated by 3D Printing and Surface Modification"

_polymers, 2024, doi:10.3390/polym16131874_

Round 1

Reviewer 1 Report

Comments and Suggestions for Authors

The paper is well-written. It studied the unidirectional wetting of liquid on patterned surfaces. However, the content is too short. Here are some suggestions for improvement.

1. Please give more detailed experiment setup for Figure 3 and 4. Such as the volume of the water droplet.

2. The study can be extended to repeated experiments or droplets. Hence, it can be useful for fog harvesting, which collects water over time.

3. More patterns could be added, or the parameters of triangular prism patterns could be modified for a complete study.

4. It can be compared to a sample without a pattern or straight channel. Hence, the effect of bioinspired surfaces can be discussed.

Author Response

We sincerely thank the reviewer for the insightful suggestions and support. 

Reviewer 2 Report

Comments and Suggestions for Authors

1. For the pattern design there is no description about the dimensions in figure 2 , why these dimensions? What bio inspiration do they relate to, it seems the link between cactus barbs and spider silk is tentative at best. i think extra work is needed to make this scientific link.

2. You say 'The optimal design for manipulating effectively the water self-transport has been investigated by adjusting the thickness (T) and the spacing (S) of the modified samples.' This needs to be explained properly in the experimental section 2.

3. All experimental variants should be made totally clear in the experimental section. Then when reading the results it will become much more clear.

4. there are no statistical results demonstrated here. it is not clear how often the experiments were repeated and without this there is no real evidence.

Comments on the Quality of English Language

A basic edit will be required.

Round 2

Reviewer 1 Report

Comments and Suggestions for Authors

the authors have addressed all comments

Author Response

Comment 1: The authors have addressed all comments.  

Response 1:  We sincerely thank the reviewer for the insightful suggestions and kind support. 

Reviewer 2 Report

Comments and Suggestions for Authors

The improvements are welcome and make the paper better.

The main problem is that there is a lack of statistical evidence. You pointed out that three repeats were conducted. This number is not enough for a scientific paper. I think that you should do the experiment with at least ten repeats and statistically validate the results.

Comments on the Quality of English Language

Fine

Author Response

Comment 1: The improvements are welcome and make the paper better.

The main problem is that there is a lack of statistical evidence. You pointed out that three repeats were conducted. This number is not enough for a scientific paper. I think that you should do the experiment with at least ten repeats and statistically validate the results.

Response 1: 

We thank the reviewer for this important comment. Unfortunately, we are not able to fabricate additional samples, conduct experiments, analyze results and revise the manuscript within a week by the due date 6/24. Also, the 3D printer Form2 we used to fabricate samples was scrapped. The current 3D printing system and materials are different from the ones we reported in the manuscript. We truly appreciate the reviewers comments and emphasis on the statistical evidence and validation of at least 10 times. We will keep this suggestion in mind in our future studies.